# Different Adaptive Potential of Norway Spruce Ecotypes in Response to Climate Change in Czech Long-Term Lowland Experiment

**Václav Šimůnek** [1,*], **Jan Stejskal** [1], **Jaroslav Čepl** [1], **Jiří Korecký** [1], **Zdeněk Vacek** [1], **Stanislav Vacek** [1], **Lukáš Bílek** [1] and **Michal Švanda** [2,3]

1   Faculty of Forestry and Wood Sciences, Czech University of Life Sciences Prague, Kamýcká 129, 165 00 Prague, Czech Republic; bilek@fld.czu.cz (L.B.)
2   Astronomical Institute of the Czech Academy of Sciences, Fričova 298, 251 65 Ondřejov, Czech Republic; svanda@sirrah.troja.mff.cuni.cz
3   Astronomical Institute, Charles University, V Holešovičkách 2, 180 00 Prague, Czech Republic
*   Correspondence: simunekv@fld.czu.cz

**Abstract:** As a result of climate change, Norway spruce (*Picea abies* [L.] Karst.) is dying across Europe. One of the primary reasons for this is the cultivation of unsuitable spruce provenances and ecotypes. This study deals with the growth and genetics of the ecotypes of Norway spruce, the most important tree species of the Czech Republic. At the study site, namely Cukrák, an experimental site was established in 1964 to plant three basic spruce ecotypes: low-elevation (LE), medium-elevation (ME) and high-elevation (HE) ecotype. A dendrometric inventory, dendrochronological sampling and genetic analyses were carried out on individual trees in 45 to 46 years old spruce stands. The ME ecotype was the most productive in terms of its carbon sequestration potential, while the HE ecotype had the lowest radial growth. All ecotypes exhibited a noteworthy negative correlation between tree-ring growth and seasonal temperature, annual temperature, previous year September to current year August temperature, June to July temperature, as well as individual monthly temperatures from previous May to current August. The relationship of annual and seasonal precipitation to growth was significant only for the LE and ME ecotypes, but precipitation from previous year September to current year August and precipitation from current June to July were the most significant for all ecotypes, where the ME had the highest r value. The HE ecotype does not adapt well to a dry climate and appears to be unsuitable compared to the other ecotypes under the studied conditions. This study also documented intra-population genetic variation within years of low growth, as evidenced by significant clonal heritability. The selection of the appropriate spruce ecotypes is essential for the stability and production of future stands and should become an important pillar of forest adaptation to climate change.

**Keywords:** *Picea abies* [L.] Karst; drought; tree-ring growth; genetic variability; carbon sequestration

## 1. Introduction

Norway spruce (*Picea abies* [L.] Karst.) is a vital commercial tree species in the Czech Republic (CR) and Central Europe [1]. At present, it accounts for 48.1% of tree species in the CR [2]. However, it has recently faced significant challenges in Central Europe, primarily due to bark beetle infestations and droughts [3]. Rising average temperatures have led to more frequent droughts, contributing to the increased dieback of Norway spruce [4,5]. The poor condition of spruce stands resulting from drought is often compounded by bark beetle infestations, further exacerbating the decline in spruce stands [6,7]. Norway spruce has a limited capacity to adapt to summer droughts due to its shallow root system, making spruce stands highly susceptible to adverse climatic events, such as droughts and windstorms [8–12]. Additionally, damage caused by browsing or bark stripping by hoofed

game animals weakens spruce stands, making them more vulnerable to climatic factors, and especially drought [13,14].

Climate change has prompted the migration of Norway spruce from lowland to middle and higher elevations. However, there is a significant dieback of spruce in non-native habitats at lower elevations in the CR [2,7,15,16]. Planting Norway spruce in these non-native habitats also weakens younger stands, making them susceptible to *Armillaria* spp. infestations [17]. Spruce stands are further impacted by the proliferation of the bark beetle *Ips typographus* and other species during warm, dry summers, which attack trees irrespective of their health status [18,19].

Due to the varying climatic conditions, the CR hosts different spruce ecotypes with diverse growth requirements. Spruce naturally thrives in a range of settings, from lower elevations to high mountain areas near the upper forest boundary. In Central Europe, three primary ecotypes of Norway spruce have evolved, each possessing distinct characteristics, genetics and adaptations to specific ecological conditions. These ecotypes are categorized based on their native distribution: the low-elevation (LE) or "lowland" ecotype (*acuminata*), found up to 500 m above sea level; the medium-elevation (ME) ecotype (*europaea*), thriving between 500 and 1000 m above sea level; and the high-elevation (HE) ecotype (*obovata*), occurring above 1000 m above sea level.

The *acuminata* ecotype stands out with its elliptical broad crown and slender, elongated branches that extend perpendicularly from the trunk. *Europaea*, on the other hand, features a primarily conical crown of medium width, with shorter and denser brush-like branches that often hang downward. Lastly, the *obovata* ecotype displays a narrow crown with short, flat branches [20–25].

The LE spruce ecotype, native to lower altitudes, thrives in cool valley floodplains with adequate soil moisture. It is characterized by a comb-shaped branching pattern, longer, wider and thicker needles, and predominantly green cones before ripening [24–28]. This ecotype necessitates a longer vegetation season and exhibits rapid growth at a younger age due to better nutrition [28]. During the air pollution crisis of the 1970s to 1990s, caused by coal power plants, this ecotype suffered the most damage from $SO_2$ emissions due to the large size of its trees and the vulnerability of its second-order vertical branches [29]. However, in recent years, this spruce type has been affected by drought and temperature extremes—the factors believed to initiate growth decline and dieback, leading to greater infestation by honey fungus and, in older stands, bark beetles [30,31]. This ecotype is more often damaged by frost [32,33].

The upland (ME) ecotype of spruce is adapted to a shorter vegetation season when compared to the lowland ecotype. It generally grows slower, has predominantly brush-like branching, a tapered trunk and a relatively slender crown, and is, therefore, better able to resist damage caused by snow and frost. In its natural habitat, this ecotype predominately forms mixed stands with silver fir and beech [30,34].

The HE spruce ecotype has the shortest vegetation season compared to the other two ecotypes, grows slowly, has generally flat, much shorter branches set perpendicular to the trunk, a considerably tapered trunk and a slender, often flag-like crown [24–26,35,36]. A red-fruited form of the cones before ripening is distinctly dominant [27]. The vegetative regeneration of spruce through layering accounts for a large proportion of the upper forest boundary ecotone [37].

The selection of the suitable ecotypes is of great importance with respect to the stability of spruce stands and their adaptation to global climate change. This has already been demonstrated during the air pollution calamity, when native mountain spruce ecotypes were better able to withstand the $SO_2$ load [29,31,38]. However, continuous changes in the environmental conditions directly influence the development and modification of physiological and morphological characteristics of woody plants [39]. The altitude, air temperature, atmospheric pressure, photoperiod, precipitation, wind speed, evapotranspiration and nutrients, as well as light gradients within stands, synergically influence the plant growth and morphological and anatomical features of woody species [40–42].

Another key determinant for the selection of the suitable individuals for the future climate is the degree of their adaptation, which is mainly determined by genetic factors [43]. The existence of clonal heritability in the increment is the first prerequisite for the future selection of resistant material. On the other hand, the relationship of heritability to drought and cyclic weather changes is still a relatively understudied phenomenon [44,45].

The optimum for the growth and production of autochthonous spruce populations is achieved in mountain spruce forests [31,46,47]. Lowland spruce ecotypes, given that they grow in habitats with the longest vegetation season, could adapt best to climate change because they are acclimated to the relatively highest average air temperature in the vegetation season and the lowest precipitation. However, this may not apply because the physiological performance of Norway spruce has declined in recent years in Europe and is currently affected by several abiotic and biotic stressors [48]. The biggest issues are drought and the expansion of various insect pests, especially bark beetle species like the spruce bark beetle [2].

This study describes the unique growth response of all the examined Czech spruce ecotypes on a low-lying clonal archive site. The primary emphasis of the study is the comparison of the tree-ring increment of each ecotype in the context of the genetic variability of the represented ecotypes and clones. Dendrochronological analysis is used for the evaluation, complemented by analyses including correlating the monthly, seasonal and annual precipitation and temperature data. The obtained tree-ring series of each ecotype are compared using standard dendrochronological methods, and then the clonal heritability is estimated for the selected tree-ring increment years. Finally, the current production potential of the compared ecotypes on the plot is also evaluated, including differences in carbon storage.

## 2. Methodology

### 2.1. Study Area

The study area, Cukrák, is located in the central part of the CR, near the capital city of Prague, on a mild northwestern slope. The ecotype research plot is found at an altitude of 320 m a.s.l. The characteristic soil type in the area is Cambisol. According to the Köppen classification, the climate is temperate oceanic /Cfb/ [49]. For a more detailed description of the site, see Table 1.

**Table 1.** Basic site and stand characteristics of the research plot.

| Plot Name | Location | Altitude (m a.s.l.) | Exposure | Slope (%) | Soils Type | Forest Type | Climate Classif. |
|---|---|---|---|---|---|---|---|
| Ecotype research plot | 49°56′22.2″ N 14°20′57.6″ E | 320 | NW | 8 | Cambisols | 2K4 | Cfb |

Notes: NW—northwest; 2K4—Fageto-Quercetum acidophilum, Festuca ovina; Climate classification according to Köppen (Köppen, 1936): Cfb—temperate oceanic climate.

In 1964, an experimental plot with Norway spruce ecotypes was established on this site. Verified ecotypes of Norway Spruce were planted in a three-year period, between 1964 and 1966. The LE ecotype seed material was collected in the Konopiště area at an altitude of 360 m a.s.l. Seed material for the ME ecotype was collected in the areas of Janov nad Nisou (736 m a.s.l.), Janovice u Rýmařova (770 m a.s.l.) and Nové Město na Moravě (775 m a.s.l.). Seed material for the HE ecotype was collected in the Vrchlabí area at an altitude of 1160 m a.s.l.

For the climatic characteristics of the locality, data from the meteorological station of Neumětely (322 m a.s.l.) were used [50]. For a detailed description, see Table 2. The period of between 1975 and 2018 was used for the assessment.

**Table 2.** Basic site climatic characteristics of study site.

| Plot | Meteo. Station Name | GPS of Meteo. Station | Station Altitude (m a.s.l.) | Distance to Plot (km) | Seasonal Period | Annual Temp. (°C) | Seasonal Temp. (°C) | Annual Prec. (mm) | Seasonal Prec. (mm) |
|---|---|---|---|---|---|---|---|---|---|
| Ecotypes study site | Neumětely | 49°51′08.3″ N 14°02′10.6″ E | 322 | 24.51 | May to September | 8.7 | 15.9 | 535 | 323 |

Notes: Meteo.—meteorological; Seasonal period is month range; Temp.—mean year air temperature from 1975–2018; Prec.—annual precipitation from 1975–2018.

### 2.2. Plant Material

This experiment encompasses a unique clonal Norway spruce common-garden trial established in the CR (49°56′22.2″ N 14°20′57.6″ E) in 1970. The vegetatively-propagated material used in this experiment is derived from multiple Norway spruce populations in the CR. Grafted trees were planted as clonal rows with ten individuals per clone in 3 m distances; the distance between rows was 6 m. The whole site was subsequently pruned so that the final spacing was 6 m × 6 m. The average tree height on the plot was 20.6 m; sd = 2.95 m, and the average DBH was 33.1 cm; sd = 7.7 cm.

What sets this trial apart is its inclusion of all morphotypes of Norway spruce found in the country. These morphological forms, referred to as putative ecotypes, exhibit distinct characteristic features corresponding to their altitudinal origins.

In terms of the genotypes' altitude origin, the LE form (*acuminata*) originated from an altitude of 360 m a.s.l., the ME form (*europaea*) from an altitude range of 770–775 m a.s.l. and the HE form (*obovata*) from an altitude range of 1145–1175 m a.s.l.

### 2.3. Research Plot Description

This experimental research plot is situated in one single studied locality in a low-relief area with an altitude ranging between 320 and 340 m a.s.l. It was established in 1970 [51]. All three of the studied ecotypes are located in the studied research plot, where individual trees are randomly distributed over the area. Basic dendromegtric data were measured on each tree in the research area. Each tree was drilled for denrochonological analysis, with all of the assessed trees being recorded in terms of their origin and distribution across the area. The bedrock primarily consists of clayey Algonkian phyllite slate, with varying thickness of loess and sloping clay overlaps. The soil in this area can be described as medium-deep Cambisols with a high content of skeletal material, exhibiting signs of reduction processes in certain locations. The upper horizons of the soil are clayey, while the lower horizons are composed of heavier, silty clay. Some areas lack a loess cover, and the soil generally has a higher proportion of coarse particles (>2 mm).

### 2.4. Data Collection

For the dendrochronological analysis, increment cores were taken from Norway spruce trees 1.3 m above the ground using an increment borer. Individuals belonging to each spruce ecotype were selected for sampling, and dendrochronological samples were taken from healthy trees of different heights from each ecotype. The height of all the sampled trees was measured using a Laser Vertex hypsometer (Haglöf, Långsele, Västernorrland, Sweden) with an accuracy of 0.1 m. Tree diameter at breast height was measured using a Mantax Blue metal calliper (Haglöf, Långsele, Västernorrland, Sweden) with an accuracy of 1 mm. A total of 76 dendrochronological samples were taken. The number of samples collected for each of the studied ecotype ranged between 20 and 31 individuals (Table 3).

**Table 3.** Characteristics of tree-ring chronologies of Norway spruce ecotypes on research plot in 1973–2018.

| Plot Name | No. Trees | Mean RW (mm) | SD RW | Mean Min–Max (mm) | Age Sample | AR1 | R-bar | EPS | SNR |
|---|---|---|---|---|---|---|---|---|---|
| LE | 31 | 3.75 | 1.79 | 1.94–5.48 | 46 | 0.54 | 0.52 | 0.96 | 26.13 |
| ME | 25 | 4.28 | 2.10 | 2.68–6.04 | 45 | 0.61 | 0.46 | 0.95 | 17.32 |
| HE | 20 | 3.27 | 1.73 | 1.49–4.74 | 46 | 0.60 | 0.33 | 0.88 | 11.51 |

Notes: No. trees—number of trees; Mean RW—mean ring width in mm; SD RW—standard deviation from ring width in mm; Mean min–max—mean ring-width range in mm from smallest to biggest tree; Age Sample—age range of sampled tree-ring time series; AR1—first-order autocorrelation; R-bar—inter-series correlation; EPS—expressed population signal; SNR—signal-to-noise ratio.

The collected cores were then recorded and measured using a LINTAB measuring table (Rinntech) and an Olympus microscope. The measuring table achieves an accuracy of 0.01 mm and the TSAP-Win software (version 4.82b2) was used to record the values of the cores [52]. Cross-dating of the cores was performed using the CDendro software (Cybis Elektronik and Data AB, Saltsjöbaden, Sweden) to achieve a cross-correlation index value greater than 25 for each sample compared to the others.

Monthly air temperature and precipitation data were provided by the Czech Hydrometeorological Institute, Prague [50]. Data on the establishment of the experimental plot were taken from the Forestry and Game Management Research Institute in Strnady [51].

*2.5. Data Analysis*

2.5.1. Dendrocrhonological Data Processing and Analysis

Dendrochronological data from the Norway spruce ecotypes were processed using the R software, using the "dplR" and "pointRes" packages [53–56]. Negatively exponential detrending with an interleaved 67% spline (31 year spline) was used to detrend individual trees, following the instructions in the "dplR" package [57]. Detrending serves to remove the age trend while preserving low-frequency growth signals and eliminates the effect of highly variable juvenile rings [58,59]. An expressed population signal (EPS) was calculated for the dendrochronological data. The EPS quantifies the reliability of a chronology by measuring the proportion of the total variance shared with an idealized infinite tree population. The limit for using dendrochronological data series for comparison with climate data was a significant EPS threshold >0.85 [57]. Dendrochronological indicators were calculated. The signal-to-noise ratio (SNR) quantifies the signal strength of a chronology, while the inter-series correlations (R-bar) measure the degree of correlation between tree-ring series within the chronology [59,60]. The first-order autocorrelation (AR1) was calculated. The EPS, SNR, R-bar and AR1 indices were calculated according to the instructions to "dplR" [61] based on general dendrochronological theories [58,59,62]. A description of the dendrochronological characteristics and data is provided in Table 3.

The analysis of pointer years based on changes in relative growth was conducted [63,64]. The negative pointer year for each tree was identified using the strict criterion, where the tree-ring increment did not reach 40% of the average increment calculated from the previous four years. The occurrence of a negative pointer year for the entire time series was determined if at least 50% of the trees in a given year exhibited a pointer year. Similarly, the positive pointer year was identified for a tree that had a 60% higher increment compared to the average increment of the preceding four-year interval. The occurrence of a positive pointer year for the entire time series was determined if 60% of the trees within a specific year class showed a higher increment. This method helps to reduce the impact of common density fluctuations within growth rings, particularly in conifers with wider rings growing in semi-arid areas [63]. The pointer years are characterized as deviations from the average tree-ring growth, expressed as a percentage, and identified based on the most frequent event within a particular year class [65].

We implemented the computation of resilience components, including resistance, recovery, resilience and relative resilience [66]. The calculation of components of resilience includes resistance (inverse of growth reduction during the episode), recovery (increased growth relative to the minimum growth during the episode), resilience (ability to return to pre-episode growth levels) and relative resilience (resilience weighted by the damage experienced during the episode). These components are determined by identifying the negative pointer year for individual spruce ecotypes.

Resistance corresponds to the ratio between the growth during the low-growth period and the growth during the respective pre-low growth period. Recovery corresponds to the ratio between the post-low growth and the growth during the respective low-growth period. This index is positive, with values up to one indicating a decline in growth after the episode. Resilience is the ratio between post-low growth and pre-low growth. Relative resilience is the resilience weighted by the damage experienced during disturbance and indicates performance before, during and after disturbance [65,66].

### 2.5.2. Tree-Ring Analysis with Precipitation and Temperature

The Pearsons correlation table was accomplished using the Statistica 13 software to determine the basic influence of temperature and precipitation on the detrended RWI and tree-ring width [67]. A basic computational analysis of the effect of monthly precipitation and temperature was performed for the period of May to September (Table 2). The sum of monthly precipitation in a given seasonal period was used to analyze the seasonal precipitation. This yearly and seasonal window was chosen for the evaluation of the direct effect on the studied ecotypes in the current year. The DendroClim 2002 software was used for the analysis of the "response and correlation" functions. This software covers individual months in the previous vegetation season for monthly temperatures and monthly precipitation. This analysis was performed from May of the previous vegetation season to September of the current vegetation season. Statistical significance was established at $p < 0.05$ for the correlation coefficients [68]. The monthly, seasonal and annual correlations can describe closer differences in the effect of precipitation and temperature on individual ecotypes. The data of precipitation and temperature from previous year September to current year August and from current June to current July were applied to describe the precipitation and temperature cumulative effect on the spruce ecotypes.

The Superposed Epoch Analysis (SEA) was computed [69]. The SEA analysis helps to mitigate the influence of anomalies during positive and negative pointer years. This statistical method is utilized to investigate periodic patterns within a time sequence or to identify correlations between two different time series. In this study, the key times were defined as positive and negative pointer years in the reference time series. Subsets of data were extracted from the second time series within a specified range around each key time. These subsets from both time series were then superimposed, aligning them at negative pointer years (1982, 1993, 2003, 2004, 2012, 2013) or positive pointer years (1977, 1997, 2011). Finally, the averaged values were calculated to facilitate inter-comparisons.

We applied the SEA technique to enhance the trends synchronized with the seasonal precipitation and seasonal temperature. The statistical significance was tested utilizing the Kruskal-Wallis test [70]. The differences are considered statistically significant when the *p*-value is below the chosen threshold ($p = 0.05$).

### 2.5.3. Heritability Data Processing

For heritability estimation, the ASReml library for R version 4 [71] was used. A univariate linear mixed model was fitted with the following terms:

$$y = 1\mu + X\beta + Zc + e \tag{1}$$

where $y$ corresponds to the increment data vector; $\mu$ is the overall mean effect; $\beta$ is the fixed effect associated with ecotype; $Z$ is the clonal effect with the Multivariate Normal (MVN) Distribution, $c \sim MVN(0, \sigma^2_c I_c)$; $e$ is the random vector of errors, with $e \sim MVN(0, R)$.

The number 1 and the letters *X* and *Z* designate a vector of ones and incidence matrices for associated fixed effects and random effects, and $I_c$ is an identity matrix of order c. *R* is a matrix of variance-covariance of errors for each field position. Two forms of *R* matrix were evaluated, one considering independent errors, i.e., $R = \sigma^2 I_n$ and another based on a separable first-order autoregressive process (AR1) in rows and columns, for which the *R* matrix is $R = \text{If}^2[\text{AR1}(p_{\text{col}}) \otimes \text{AR1}(p_{\text{row}})]$; where $\text{If}^2$ is the residual variance, and $\text{AR1}(p_{\text{col}})$ and $\text{AR1}(p_{\text{row}})$ represent a first-order autoregressive correlation matrix. Several models were fitted, with consideration of independent or autoregressive errors. The best model was selected based on likelihood ratio tests (LRT) and approximated F-tests [72].

Broad-sense heritability ($H^2$) was estimated for each response variable based on the following formula: $H^2 = \sigma^2_c/(\sigma^2_c + \sigma^2)$, where $\sigma^2_c$ is the genetic variation attributed to represented clones; approximated standard errors were obtained using the Delta method.

### 2.5.4. Stand Structure and Biomass Analysis

The SIBYLA Triquetra 10 alpha software was utilized to assess the fundamental production characteristics of the spruce trees [73]. The volume of trees was determined using volume equations [74]. The slenderness coefficient was calculated as the height-to-diameter at breast height (DBH) ratio [75]. The carbon (C) content in tree biomass was calculated using the method outlined in Bublinec, 1994 [76]. The above-ground tree biomass, including stems, branches and needles, was estimated [74,77–79]. The biomass of tree roots was calculated using a model by Drexhage and Colin, 2001 [80].

Statistical analyses of tree production parameters, such as DBH, height (HDR-height to DBH ratio), crown length, stem volume and tree carbon storage, among different ecotypes, were conducted using the Statistica 13 software (StatSoft, Tulsa, USA). The data were initially tested for normality using the Shapiro-Wilk test, and then the Bartlett variance test [81,82].

When the assumptions of normality and homogeneity of variances were met, the differences between the examined parameters were tested using a one-way analysis of variance (ANOVA) followed by the Tukey's Honest Significant Difference test [83]. If the assumptions of normality and homogeneity of variances were not met, the investigated characteristics were tested using the nonparametric Kruskal-Wallis test [70]. Multiple comparisons after the Kruskal-Wallis test were performed using a method by Siegel and Castellan [84].

## 3. Results

### 3.1. Tree-Ring Growth of Norway Spruce Ecotypes

The tree-ring increment of Norway spruce (Figure 1) shows that between 1973 and 2018, there is a visible difference between the different ecotypes. The ME ecotype predominantly exhibits the highest ring width increment, while, conversely, the lowest increment is observed in the HE ecotype. The LE ecotype is similar to the increment of the HE ecotype; however, notably, this ecotype is rather average in growth, primarily at the beginning of growth until 1978. Since 1978, the ring widths between the LE and ME ecotypes are quite similar, and their growth is almost identical.

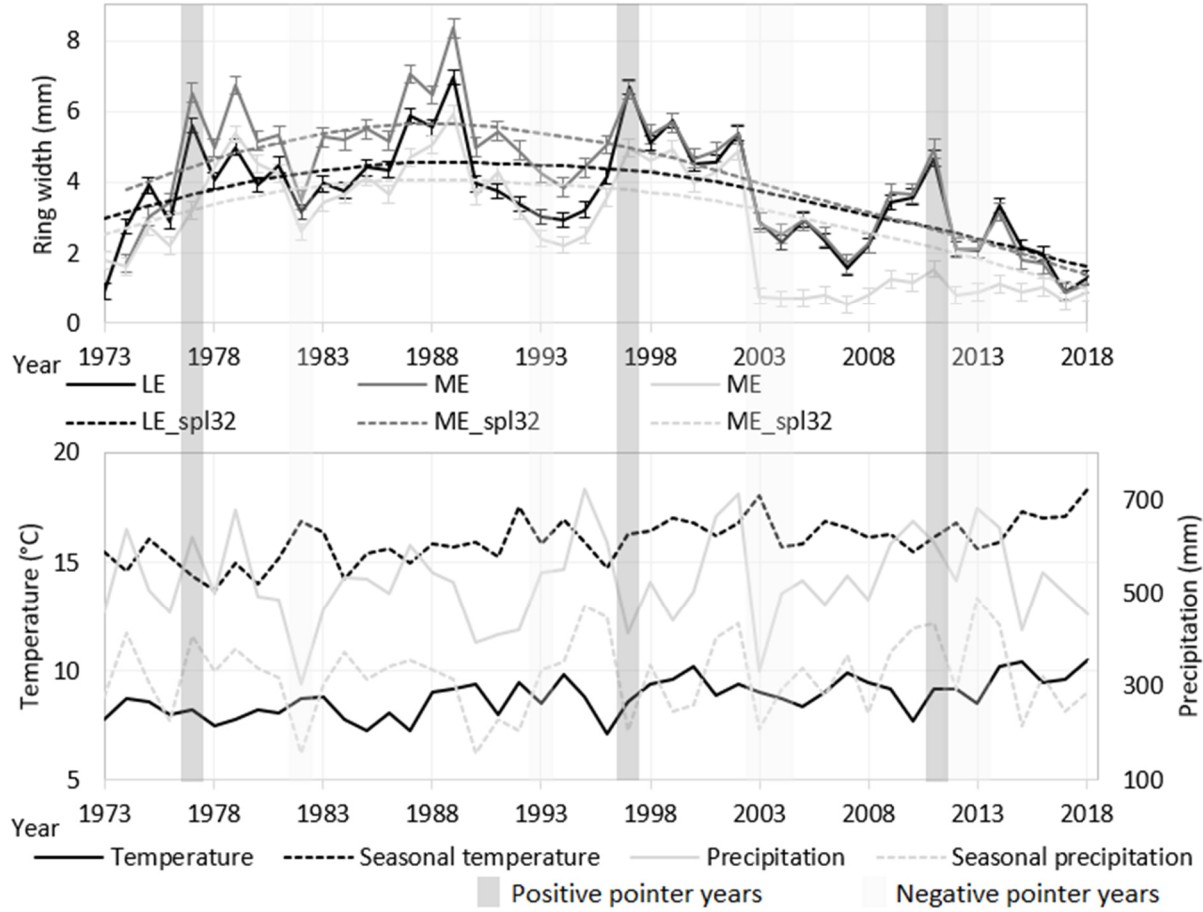

**Figure 1.** Ring-width increment of Norway spruce ecotypes on research plot (**up**) in years from 1973–2018 compared with seasonal and annual datasets of temperature and precipitation (**down**) with most common positive (dark grey) and negative (light grey) pointer year period colored area; spl32 describes 32-year data interpolation; the error bars of tree-ring width show the one-standard-error half-widths.

Regarding the pointer years (Figures 1 and 2), the greatest positive tree-ring growth is found at the intersection of 1977, 1997 and 2011, when the ecotypes of LE and ME showed the greatest growth. In contrast, the lowest growth in the positive periods was continually recorded for the LE ecotypes. In the context of the climatic data analysis, it was frequently observed that there was coincidence during positive pointer years, where the temperatures typically exhibited a slightly higher values, while the precipitation levels remained relatively stable or slightly increased. In terms of the deviation of the percentage increase, the positive pointer years (Figure 2) were most stable for the LE ecotype, with three stable positive years: 1977, 1997 and 2011. The ME ecotype showed only one pointer year (1997). The LE ecotype showed a significant increase in the ring width at the beginning of the time series of tree-ring curves, with pointer years recorded in 1975, 1976, 1977 and 1997. Nevertheless, even though there were four recorded positive pointer years, the low elevation ecotype displayed the smallest growth increment during these years.

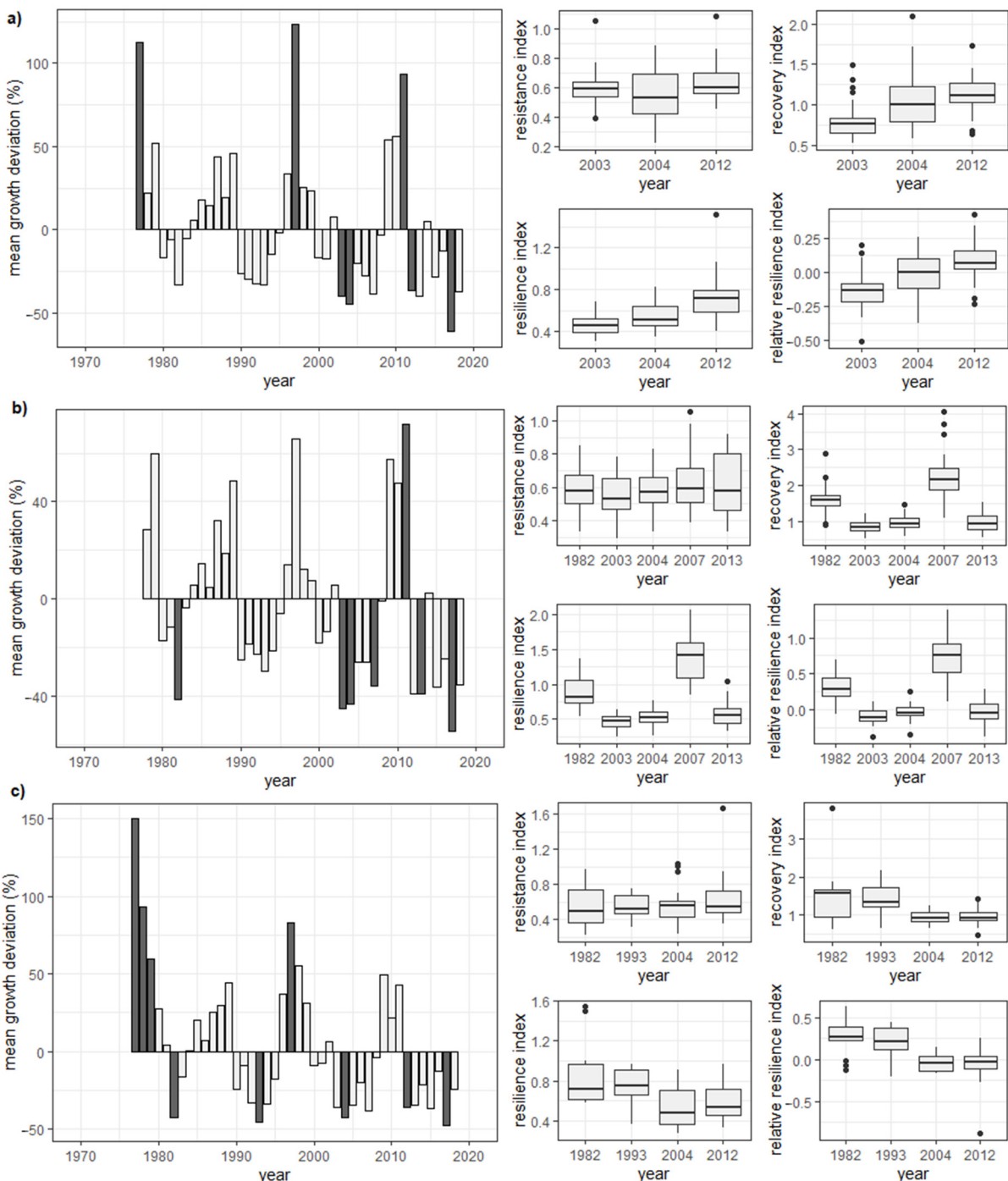

**Figure 2.** Bar plot of mean growth deviation plotted in 1977–2018 with the pointer years highlighted in dark grey (**left** panel) with indices of resistance, recovery, resilience and relative resilience (**right** panel) during the negative pointer years. The ecotypes of LE (**a**); ME (**b**); and HE (**c**); the dot—outlier.

Negative pointer years are generally recorded in ring-width declines in 1982, 1993, 2003, 2004, 2012 and 2013. During the negative pointer years, the growth declined in all of the studied ecotypes, yet it was the HE variant that recorded the lowest ring width. However, the ME variant recorded the highest number of negative pointer years for 1982, 2003, 2004, 2007, 2012 and 2017. The LE variant recorded negative pointer years predominately in 2003, 2004, 2012 and 2017. In contrast, the HE variant was stressed steadily over the long-term throughout the growth years 1982, 1993, 2004, 2012 and 2017, with a negative pointer year occurring approximately every ten years.

The most pronounced effect of the influence of warm years and a lack of precipitation has been observed since 2003, when all ecotype variants experienced a significant decline in tree-ring growth. That year's largest long-term negative effect was on the HE ecotype, which failed to match the ring-width increment of the LE and ME ecotypes. Thus, since 2003, the HE ecotype has lagged behind the other two ecotypes, averaging, for example, 1.7 mm in 2011, while the other two ecotypes grew up to 5 mm in that year.

The indicators of resistance, recovery, resilience and relative resilience (Figure 2, left panel) in negative pointer years show that the highest number of negative years is recorded in the ME ecotype and the least in the LE ecotype. In terms of the resilience index, all of the ecotypes exhibit values ranging between 0.5 and 0.6, signifying that the tree-ring increment in spruce is not entirely impervious to stress. These indexes demonstrate a relatively strong ability to adapt and sustain their growth activity. The recovery index results show that the tree-ring increment of the HE ecotype was able to recover relatively well from the negative pointer years in 1982 and 1993, but has lower values—around the value of 1—in subsequent pointer years, with a negative index trend. The recovery index shows a positive trend for the LE ecotype, while the HE ecotype displays a negative trend in its recovery ability. Meanwhile, the ME ecotype maintains a relatively stable trend in the recovery index.

The resilience Index and the relative resilience index show similar results to the previous recovery index, but these indices better incorporate the ability to return the tree-ring growth to the period before the negative event. The resilience index shows an upward trend for the LE ecotype and a relatively stable trend for the ME ecotype. In contrast, the HE ecotype has resilience indices in a negative trend that has a lesser ability to return to the original growth rate.

### 3.2. Temperature and Precipitation in Comparison to Ecotype Tree-Ring Growth

The SEA analysis in Figure 3 shows the average values of the ring-width increment at the relative point of the negative or positive pointer year, defined as superposed epoch 0. The SEA results show that in the positive pointer year, the ME ecotype has the highest average increment (6 mm), growing more than the LE ecotype (5.8 mm) by a marginal difference. The lowest values during the positive pointer year are recorded for the HE ecotype, which has a tree-ring width of up to 3.5 mm. Despite being a positive pointer year for the HE ecotype, there is no notable increase in the tree-ring width, whereas the ME and LE ecotypes exhibit much higher growth. The SEA in the positive pointer years further demonstrates that in the year prior to positive growth, the average seasonal temperatures are the lowest. The (annual) temperature in the positive pointer year is 7.8 °C, but the average temperature is about 8.8 °C. For seasonal temperatures, it is 15.2 °C, but the average is 15.8 °C. The precipitation totals are at average values during positive pointer years.

During negative pointer years, the SEA analysis shows that almost all of the ecotypes respond to the negative event with similar ring-width growth. The trend is similar, with the values set at different levels in terms of the magnitude of the ring-width growth. The highest values during the negative pointer year are again, according to SEA, for the ME ecotype (3 mm), followed, with a non-significant difference, the LE ecotype (2.6 mm). The lowest values, according to the SEA, during negative pointer years are recorded for the HE ecotype (1.5 mm). According to the SEA, one relative year before the pointer year (superposed epoch −1), higher seasonal temperatures occurred, rising from an average of 15.9 °C to 16.9 °C, and the slightly above-average temperature of 16.7 °C occurred during epoch 0. There is a substantial drop in the annual and seasonal precipitation for negative pointer years. The seasonal precipitation decreased from an average of 350 mm to 300 mm in superposed epoch 0 and from 550 mm to 490 mm in annual precipitation.

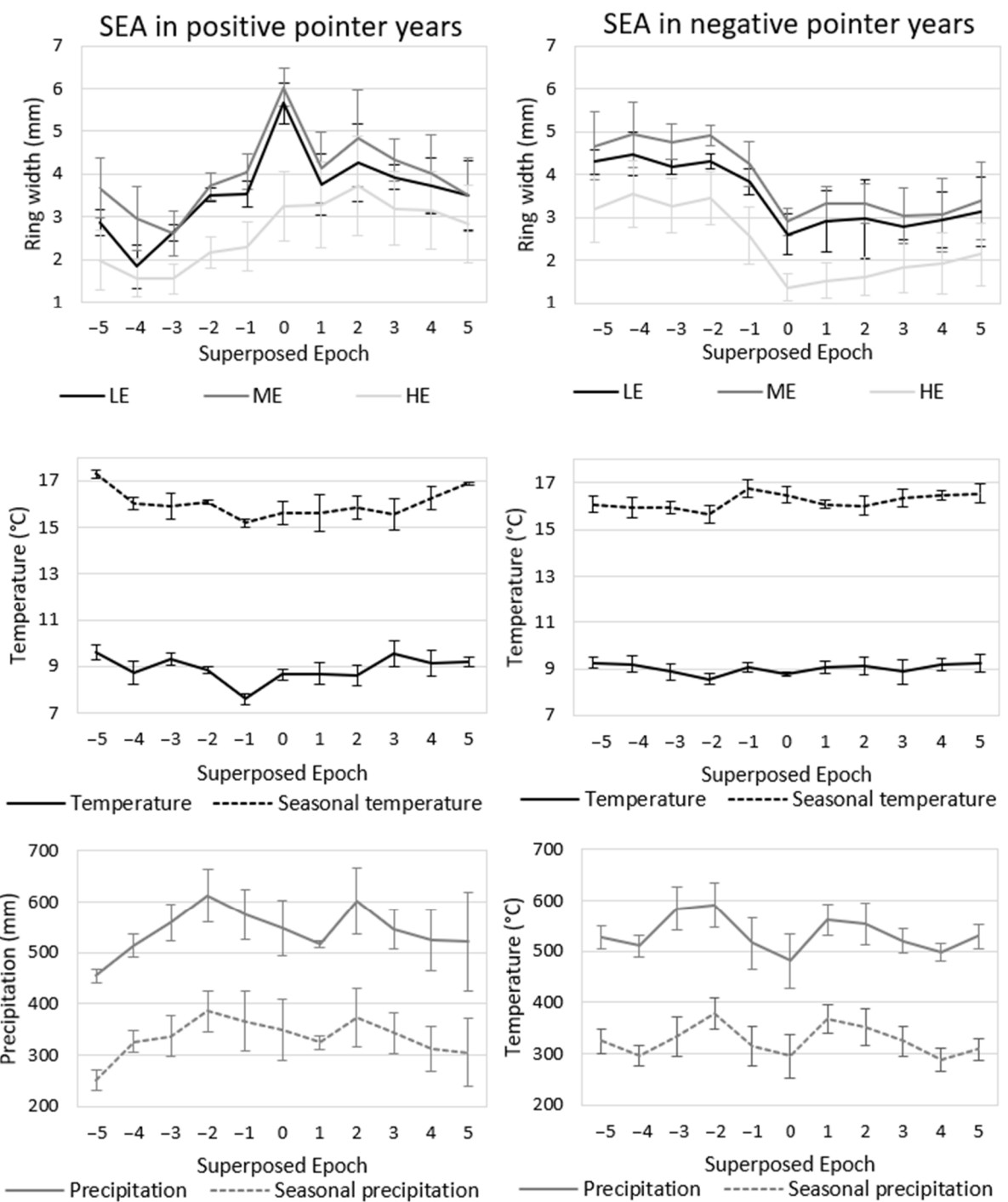

**Figure 3.** Plots of superposed epoch subsets (SEA) from positive pointer years (**left** graph panel) and negative pointer years (**right** graph panel) in relative −5 years before (**left**) and 5 years (**right**). The superposed epoch is determined in positive pointer years (1977, 1997, 2011) or negative pointer years (1982, 1993, 2003, 2004, 2012, 2013). The SEA analysis was made for ring width, temperature and precipitation. The solid lines indicate the mean superposed ring width in the reference for time series of temperature and precipitation; epoch-superposed subsets; error bars show the one-standard-deviation half-widths.

The correlation coefficients in Table 4 show that the most significant ($p < 0.05$) negative values are found for the correlations of the undetrended ring-width increment of the Norway spruce ecotypes with all types of temperature. In contrast, the detrended ring-width increment (RWI) shows only minimal non-significant negative correlations with the

temperatures. Overall, the temperature in the current June and July are correlated with the highest r, followed by the seasonal temperature. Of the research plots, the HE ecotype has the highest significant correlation ($p < 0.05$) with the temperature in the current June to July (r = −0.50), followed by the ME ecotype (r = −0.48), and the LE ecotype (r = −0.45) shows the lowest significant correlation with the temperature in June to July. The ME ecotype has the second highest significant correlation ($p < 0.05$) with seasonal temperature (r = −0.43), followed by the high-elevation ecotype (r = −0.39), and the low-elevation ecotype (r = −0.35) shows the lowest correlation.

**Table 4.** Correlation coefficients of Norway spruce ecotypes tree-ring growth and detrended ring-width index (RWI) with precipitation and temperature in 1973–2018, significant results at $p < 0.05$ are in bold.

| | Prec. Annual | Seasonal Prec. | Prec. P9-C8 | Prec. C6-7 | Temp. Annual | Seasonal Temp. | Temp. P9-C8 | Temp. C6-7 |
|---|---|---|---|---|---|---|---|---|
| LE ring width | 0.14 | 0.11 | 0.19 | 0.17 | **−0.32** | **−0.35** | **−0.35** | **−0.45** |
| ME ring width | 0.11 | 0.09 | 0.15 | 0.15 | **−0.43** | **−0.43** | **−0.43** | **−0.48** |
| HE ring width | 0.03 | −0.01 | 0.06 | 0.05 | **−0.36** | **−0.39** | **−0.35** | **−0.50** |
| LE RWI | **0.30** | 0.28 | **0.35** | **0.34** | −0.05 | −0.16 | −0.15 | −0.27 |
| ME RWI | **0.37** | **0.34** | **0.39** | **0.39** | −0.06 | −0.18 | −0.15 | −0.23 |
| HE RWI | 0.20 | 0.16 | **0.30** | **0.35** | −0.07 | −0.13 | −0.15 | −0.26 |

Notes: prec.—precipitation; temp.—temperature; P9-C8—data from previous year September to current year August; C6-7—data from current June to current July.

Annual precipitation is more positively correlated with spruce ecotypes than temperature, but the most significantly correlated ($p < 0.05$) is the detrended RWI. The ME ecotype is most significantly correlated with the precipitation from the previous September to current August and with the precipitation in the current June to July (in both cases, r = −0.39 at $p < 0.05$), but less significantly correlated with the seasonal precipitation (r = −0.34). Furthermore, the LE ecotype is significantly ($p < 0.05$) positively correlated with the precipitation (r = 0.30). The cumulative effect of precipitation on the tree-rings of all three ecotypes is significantly corelated with the precipitation from the previous September to current August and with the precipitation in the current June to July.

Of the studied ecotypes, the best correlation with temperature and precipitation is the ring width of the ME ecotype, which, coincidentally, also shows the highest increments. The exception is in the temperature in June to July, where the HE ecotype indicates the highest negative correlation with the ring width.

Figure 4 shows that the monthly temperatures correlate significantly ($p < 0.05$) more frequently and regularly than the monthly precipitation totals. The course of the monthly temperatures during the year is significantly negatively correlated for almost all months, except the winter months of the current relative vegetation season (January, February, and March). The highest correlations for temperature are observed predominately for the June and July of the current vegetation season. Among the ecotypes, the HE one is the most correlated with the monthly temperatures, mainly in the previous vegetation season.

The monthly precipitation correlates with the year-class RWI growth in July of the current vegetation season. In this month, the RWI of all the ecotypes are positively significantly correlated ($p < 0.05$) with the monthly precipitation, with the highest being the ME ecotype (r = 0.45), followed by the LE ecotype (r = 0.44), and the lowest being the HE ecotype (r = 0.41).

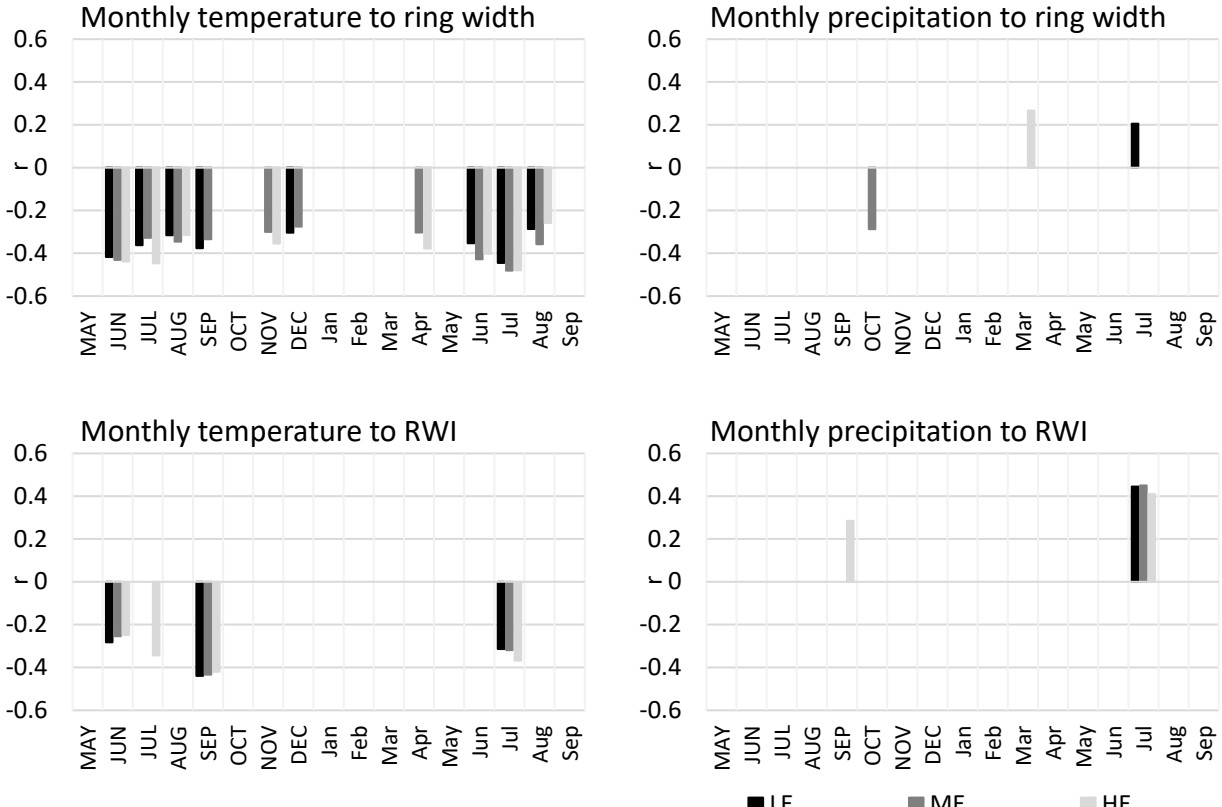

**Figure 4.** The values of significant correlation coefficients of ring-width chronology and detrended ring-width index (RWI) of Norway spruce ecotypes with the monthly temperature and monthly precipitation from May of the relative preceding year (in caps) to September of the current relative year for the period of 1975–2018. Values are statistically significant (*p* = 0.05).

Seasonal temperatures have the most substantial long-term impact on the growth of all ecotypes, as seen in Table 4 and Figure 4. The strong negative correlations indicate that lower average temperatures (Figure 4) are typically observed the year before a high RWI, while higher temperatures occur the year before a low RWI.

Precipitation totals have a less apparent connection to pointer tree-ring growth. However, when there is a significant negative pointer year, there is a sudden and notable decrease in the precipitation totals. Additionally, the results demonstrate that the correlation between precipitation and tree-ring growth (Tables 2 and 4 and Figure 4) follows a similar pattern to the magnitude of tree-ring growth for each ecotype, with the ME ecotype showing the most substantial gain, followed by the LE and the HE ecotypes, which exhibit the smallest increase. This precipitation-tree-ring growth correlation aligns with this sequence.

### 3.3. Clonal Heritability and Ecotypic Variation

The comparison of the RWI of individual ecotypes in the context of the genetic variability of the represented ecotypes and clones can be clearly seen in Figure 5.

Clonal heritability was estimated using the mixed linear model for all of the years of increment, but reaches significance only in a few selected years, and the corresponding estimation errors are variable, based on which the confidence interval (±SE × 1.96) is reported and visualized (Figure 5). It is worth noting that, with two exceptions (1982, 2013), a significant heritability of increment occurs outside the pointer years. Moreover, the two pointer years mentioned are both negative. Further, the graph shows that 10 of the 13 significant pointer year increments occurred after 2003, i.e., at relatively older stand ages.

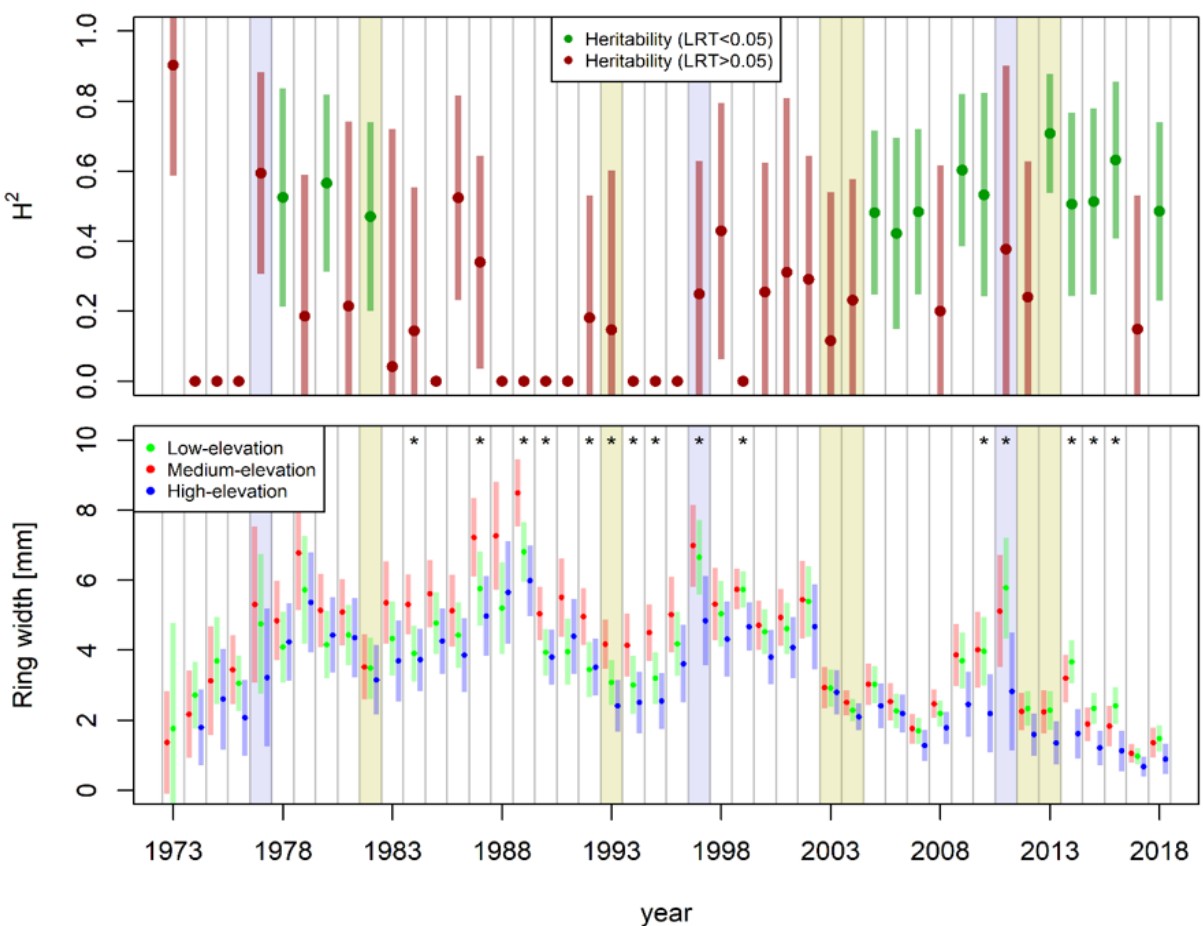

**Figure 5.** The clonal heritability with 95% confidence interval in green when logLikelihood test < 0.05 and in red if logLikelihood test > 0.05; bottom part: ecotypic variation plotted as ecotypic mean with a 95% confidence interval for each ecotype; asterisks at the top indicate ANOVA-style Wald test of factor ecotype < 0.05 (the symbol * is a statistical significant value for heritability in the year); individual years are separated with vertical lines with most common positive (blue) and negative (beige) pointer year period colored area.

Ecotypic variability is another phenomenon related to the original local adaptation of the populations represented in the test from different altitudes. Like clonal heritability, the ecotypic variability was described based on a mixed linear model. Compared to clonal heritability, the significant difference between ecotypes falls more often in the most common pointer years, both positive (1977, 1997, 2011) and negative (1993). However, in negative years, the drops in heritability are insignificant values. Another difference lies in dating the significant ecotypic variability more frequently in the early to middle phase of stand development between 1984 and 1999. The overlap of significant ecotypic variability and clonal heritability for the annual increment is rare and falls into the recent past.

Based on this analysis, the performance and ranking of the ecotypes are almost identical to those in Section 3.1., with the montane ecotype consistently performing the worst in terms of increment.

### 3.4. Production Potential

In terms of the production of individual ecotypes, a significant ($p < 0.001$–$0.05$) difference was found between all of the dendrometric parameters examined, with the exception of the HDR ($F_{(2, 68)} = 1.50$, $p = 0.23$; Figure 6). The significantly ($F_{(2, 68)} = 9.68$, $p < 0.001$) highest difference between ecotypes was found in the DBH, with the highest dimension reached by the ME ecotype (35.5 cm ± 7.0 SD) and the lowest diameter by the HE eco-

type (19.6 cm $\pm$ 3.5 SD). Likewise, the significantly (KW = 6.76, df = 2, $p < 0.05$) most prominent height was achieved by the ME ecotype (21.2 m $\pm$ 2.5 SD) and the LE ecotype (20.7 m $\pm$ 1.7 SD) when compared to the HE ecotype (13.5 m $\pm$ 4.5 SD). The stem volume ranged between 0.176 m$^3$ (HE ecotype) and 0.875 m$^3$ (ME ecotype). The carbon sequestration was 6-fold higher in the ME ecotype (142.1 kg tree$^{-1}$) compared to the HE ecotype (25.6 kg tree$^{-1}$; F$_{(2, 68)}$ = 6.1896, $p < 0.01$).

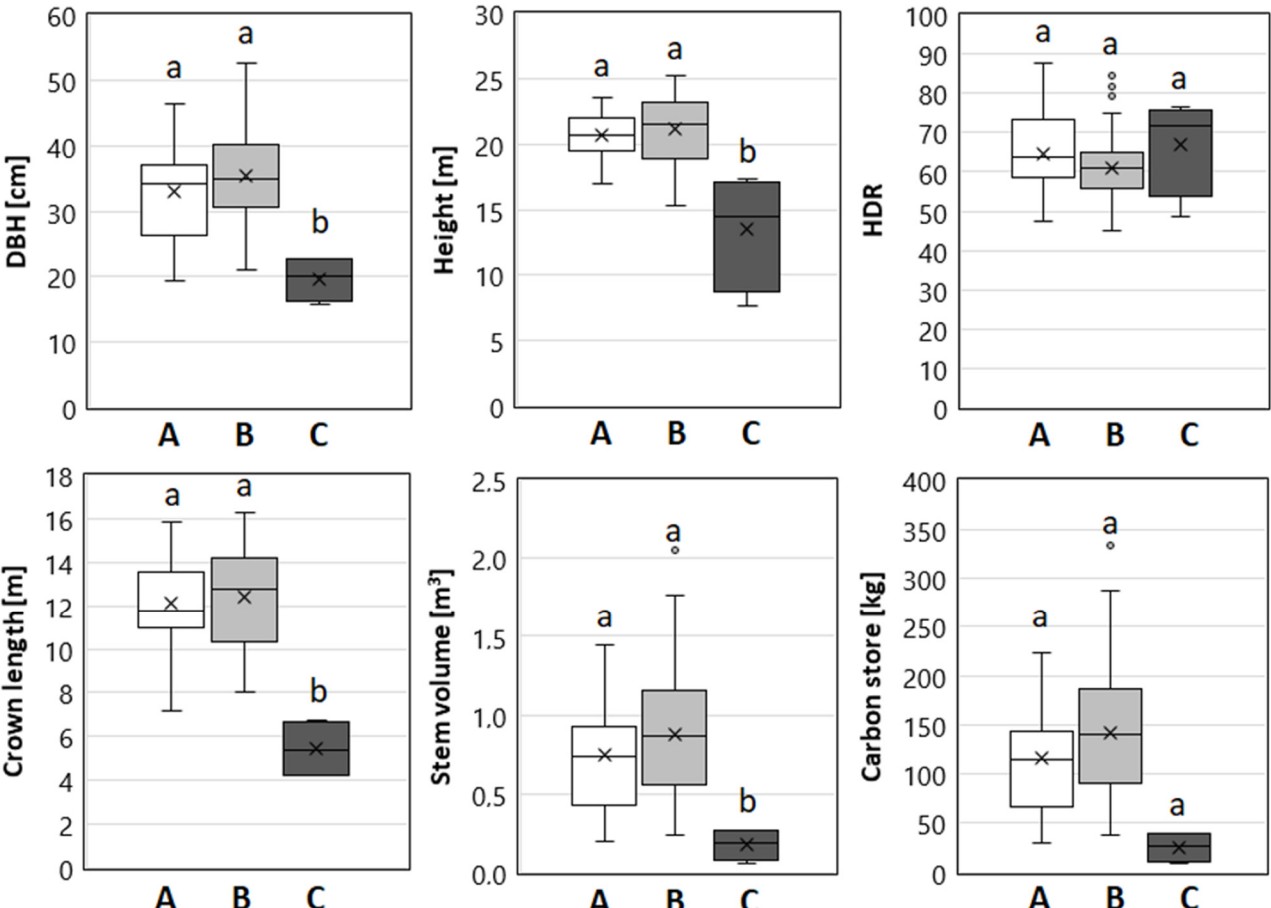

**Figure 6.** The differences between spruce ecotypes (A—LE, B—ME and C—HE) in DBH, tree height, HDR, crown length, tree stem volume and tree carbon storage; Indices above the variants in boxplots depict statistically significant differences (significantly different variants are marked with a different letter), the symbol °—outlier.

## 4. Discussion

### 4.1. Limitations

The advantage of this study is that the experimental plot was established in 1964, using clones of spruce trees with traceable origins and morphotypes that were typical and natural to the original site. Each origin site was evaluated for its relative ecotype at that time. The advantage of the experiment was that it maintained a relatively high number of spruce individuals until 2018, despite the high mortality of spruce stands in the vicinity of the experimental plot. A limitation of this experimental design is that there is only one site with such comprehensive data on the origin of spruce ecotypes in the CR; therefore, the experimental design is limited to comparing Norway spruce ecotypes within a single site. Furthermore, it must be mentioned that the reported clones are grafted and the heterogeneity of rootstocks may introduce unwanted variability in clonal replications, which cannot be mitigated statistically.

### 4.2. Growth Conditions and Ecotype Response of Norway Spruce on the Climate

Naturally occurring spruce at lower altitudes (also lowland ecotype) are characterized by high productivity when compared to other tree species, such as hornbeam, oak, beech or large-leaved lime. However, spruce show a high sensitivity to the synergy of low precipitation with the combination of high temperatures, which appeared to be a limiting factor for this tree species in recent years [85]. The results from Figures 1 and 3 show that significant drought manifestations have recently been occurring through synchronous increases in the temperature and a decrease in precipitation during negatively significant years, with the largest decrease recorded in 2003 for all spruce ecotypes. As a result of the higher negative impact of drought in the lower and middle elevations of Central Europe, spruce stands are in significant decline [7].

The exception to this decline was the LE ecotype of Norway spruce, which grows naturally in cool and moist valley floodplains of lower to middle elevations [86]. The results of the tree-ring analyses show that the LE ecotype is just behind the ring-width increment of the ME ecotype. According to the correlations of the tree-ring growth to all of the studied temperature variants (Table 4), the LE ecotype has the lowest significant correlation coefficients ($p < 0.05$) of all the studied ecotypes. In forestry, it is generally believed that the LE ecotype is best adapted ecologically to drier climates, which is also confirmed by the results of our data [30]. The cumulative effect on the growth of spruce ecotypes in the previous year September to current year August was higher, which indicates that, from the previous season to the current one, moisture is an important factor for spruce in lowlands, and from the ecotypes, the HE indicated lower coefficients, while the precipitation from June to July are the lowest for LE.

In contrast, the HE ecotype is better adapted to the colder, frosty climate of mountain and alpine regions, where winter low temperatures occur, while there is ample moisture in the upper parts of the mountains throughout the year [87,88]. However, when the HE ecotype is transferred to the lowlands, it appears that the spruce HE ecotype is more likely to have adapted to different growing conditions during the vegetation season, with the highest correlation with temperatures recorded (Figure 4) during the month of July of the current growing season. The correlations from Table 4 also show the highest significant r values of the HE ring width with the temperature of the current June to July compared to the other ecotypes. Therefore, we can hypothesize that the HE spruce ecotype grows later in the year and, thus, its need for more moisture may be evident at a time when less precipitation occurs in the lowlands, indicating the lowest correlations during the month of July with precipitation compared to the other ecotypes. Studies have confirmed that HE spruce shoots for a longer time and are therefore more resistant to frosts early in the vegetation season [89,90]. The low increment of the HE ecotype and the greater and longer-term deteriorating sensitivity in terms of declining trends in the resistance, resilience and recovery indices during the pointer year compared to the other ecotypes may also be due to the lower needle density, together with the smaller branching area, which therefore gives this ecotype a smaller area of assimilatory organs [24,91].

The ME ecotype appears to be the most productive ecotype of all the variants, having the highest average annual ring-width increment (Table 3) and the highest increments during negative and positive pointer years, in which it also shows higher relative resistance (Figure 3). In terms of the current forestry situation, spruce stands at lower and middle elevations are most vulnerable to drought, which is also evident in their increment [92]. At the same time, there is a decline in the annual growth of spruce trees in the middle and lower elevations; however, in the mountains, the ring-width growth has recently started to increase with the onset of climate change [93]. This is confirmed by our results, which recorded a large growth drop for all ecotypes since 2003, with the largest drop in growth being recorded for 2017, when there was a significant dry period, with the onset of wind calamities in 2018 culminating in the formation of salvage logging until 2021 [2,94].

The emergence of negative events in recent years has led to spruce trees beginning to retreat from the lowlands and midlands [7,95]. An explanation for this retreat may be the

surface root system of spruce, which does not allow the use of moisture from lower parts of the soil profile [11,96,97]. Furthermore, spruce also exhibit poorer wood density and strength associated with higher stomatal conductance, which is evident in diseased and weakened individuals, whose numbers increase with ongoing dry years [98]. However, in the past, spruce have been largely planted at lower elevations by forest management that relied on increasing spruce production by decreasing the elevation during different climatic conditions [99]. This is supported by the fact that spruce stands grow wider tree rings more frequently at lower altitudes, depending heavily on the habitat suitability and stand conditions of individual spruce ecotypes [31,100,101].

*4.3. Clonal Heritability of Annual Ring Widths*

The trend of heritability in the magnitude of increments observed through the years shows a significant heritability at a young age, then a roughly 20-year period when heritability in the increments did not manifest whatsoever. In contrast, since 2005, clonal heritability has increased, holding at an average level of 0.5 with an SE of roughly 0.1. The highest heritability value (0.7) ever recorded (for all years observed), with the lowest estimation error, was observed in the negative pointer year of 2013. This high value of heritability is caused by the high variability in the tree rings between the provenances of clones and ecotypes, when, in this period, the dieback effect on spruce stands is also significantly manifested.

The SEA results indicate a predominant effect of temperature on both positive and negative pointer years, but the effect of precipitation on the increments is negligible outside of negative years, which are caused by a deficit in seasonal precipitation. A paucity of scientific studies has addressed provenance-level resilience in this context, with only one pilot study on European larch; moreover, the work shows that climatic influences and genetic variability are influenced by a wide refugial origin [102]. A heritability study of Scots pine seedlings shows that dry seasons will be a selection factor for the future [44]. The results of these studies showed that the difference between the control and drought treatment in the parameter of interest may exhibit even higher heritability than the originally measured trait.

The heritability of the tree-ring parameters was calculated primarily for the wood density and other commercially important parameters. For example, in half-sib progeny trials of *Pinus radiata*, the ring-by-ring analysis of wood density parameters showed high heritability close to the pitch, which decreased towards the outside [103], which is inconsistent with our findings. In contrast, the provenance progeny test for *Picea glauca* demonstrates an increasing trend in the heritability of several individual tree-ring parameters towards the outside of the tree [104]. We suppose that the fast-growing *Pinus radiata* has different growth dynamics from spruces, as well as a different level of heritability of growth at the juvenile stage. Another factor may be the effect of drought stress in the study area, which probably peaked in the last decade. It is the ecotypic diversity of the ring-width increments that may be influenced by this stress [22].

## 5. Conclusions

The ME ecotype showed the greatest ring-width increment in the spruce, which grew slightly more than the LE ecotype. In the tree-ring increment series, the effect of drought during negative pointer years on all spruce ecotypes was evident, which was reflected in a decrease in precipitation, especially during the vegetation season. The HE ecotype exhibited the lowest ring-width increment during negative and positive pointer years, failing to capitalize on the favorable growing conditions observed in the other ecotypes. Lower mean annual and seasonal temperatures were noted one vegetation season before a significant ring-width increment in all of the ecotypes. All of the ecotypes had a significant negative correlation between ring-width increment and temperature, while the relationship with precipitation was significant only for the LE and ME ecotypes with annual and seasonal data. The monthly temperature data were crucial for the tree-ring series of ecotypes during

the vegetation season, while precipitation showed a positive correlation in July of the current relative season. The results of this experiment with planting different spruce ecotypes in the 1970s suggest that the medium-elevation ecotype is the most productive, but only by a small margin, over the LE ecotype. The HE ecotype is the weakest variant, as it grows the least in all cases, especially in the last two decades, and responds the worst to climatic fluctuations. The HE ecotype also shows a downward trend in the resistance, resilience and recovery indices during the pointer year, while the LE ecotype has the opposite increase in these indices (copes better with climatic fluctuations). The correct choice of the location of LE and ME spruce ecotypes can help forestry to adapt to climatic changes in the mid-elevations, while the HE ecotype does not adapt well to a dry climate and appears to be unsuitable relative to the other ecotypes in the studied conditions.

**Author Contributions:** V.Š. initiated and designed the study and its methodology, collected samples, analyzed the data, wrote and edited the first draft of the manuscript. J.Č. initiated and designed the study, collected samples, analyzed the data and wrote and edited the first draft of the manuscript. J.S. and J.K. initiated, designed the study, collected samples and edited the first draft of the manuscript. Z.V. analyzed the data and edited the first draft of the manuscript. S.V. contributed to the first draft of the manuscript. L.B. contributed to the first draft of the manuscript and solved financing. M.Š. analyzed the data and edited the first draft of the manuscript. All authors have read and agreed to the published version of the manuscript.

**Funding:** This study was supported by the Ministry of Agriculture of the Czech Republic, National Agency of Agriculture Research, Project No. QK21010198, Adaptation of forestry for sustainable use of natural resources, and Project No. QK1910480, Development of integrated modern and innovative diagnostic and protection methods of spruce stands with the use of semiochemicals and methods of molecular biology.

**Data Availability Statement:** Monthly temperature and precipitation data for this study were sourced from the Czech Hydrometeorological Institute (CHMU). Data on provenances and eco-types were provided by the Forestry and Game Management Research Institute (VÚLHM).

**Acknowledgments:** We would like to express our gratitude to VÚLHM for establishing and maintaining this unique trial and to Vladimír Hynek for his guidance. We would like to thank Richard Lee Manore, a native speaker, and Jitka Šišáková, an expert in the field, for checking English. We also acknowledge the Czech Hydrometeorological Institute of the Czech Republic.

**Conflicts of Interest:** The authors declare that they have no known competing financial interests or personal relationships that could have appeared to influence the work reported in this paper.

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
