# Peer review of "Different Adaptive Potential of Norway Spruce Ecotypes in Response to Climate Change in Czech Long-Term Lowland Experiment"

_forests, doi:10.3390/f14091922_

Round 1

Reviewer 1 Report

Review of the article: forests-2592070 entitled Different adaptive potential of Norway spruce ecotypes in response to climate change in Czech long-term lowland experiment

The Authors evaluated the adaptive potential of Norway spruce trees from the Czech Republic in response to climate change by applying dendrometric inventory, dendrochronological sampling, and genetic analyses.

The Authors revealed negative correlations between tree growth and air temperature from three ecotypes (low, middle and high elevations). This is interesting funding and the period and months should be mentioned in the abstract as well as possible reasons for such correlations.

The Authors found that the 3 studies' ecotypes significantly differ from each other. The Conclusion section looks like a discussion and should be shortened. Please provide the main conclusion and outcome of the study.

Figures: should be revised according to the Journal guidelines.

The paper can be of potential interest to the Forests Journal readers, but in its current form should be restructured and validated for the time periods and analyses used. My suggestions and details are below.

Abstract: Please refer to a period for which correlations with climatic parameters were revealed.

Please indicate the age of the trees used for analysis already in the abstract.

In general, the abstract should be shortened. Abbreviations for ecotypes lo elevation (EL), medium- elevation (ME) and high-elevation (HE) could be helpful.

Introduction

It is mentioned that around the 1970s to 1990s air pollution calamity.. But it is a bit unclear from which source the SO2 come from? Industry near by? Please clarify in one sentence before.

L. 61-95 mixture of study site description. Can be separated into intro and materials and methods parts?

Minor comments

L. 41 replace is in CR 48.1 to is 48.1% in CR.

L.47, L. 63,  please provide the citation in [].

L. 56 either CR or the full name everywhere

Material and Methods

L. 141 it is unclear what was collected. Please clarify about sampling and sampling design.

2.3 Research plot description

L. 165 it is unclear about which research plot the Authors talk about. Please clarify. Is it related to the common trial above or it is different? It is very confusing. It might be helpful to clarify the following:

-dendrometric inventory

- dendrochronological sampling

- genetic analysis

Please describe study sites accordingly and the plant material collected for each analysis and ecotype.

2.4 Data collection topic is incorrect here. “Data collection” – if the Authors reuse data or collect processed data for their study. My suggestion:  Sampling if the Authors collected samples for dendrochronological analysis or “Dendrochronological sampling”.

L. 175 increment borer

L. 253 “…monthly precipitation and temperature was  performed for the period from May to September”. Could the Authors explain why they selected this seasonal window? Why not September of the previous year to August of the current one? Otherwise, there is an overlap that can result in unreasonable correlations or even overestimated correlations.

L. 257-259 Could the Authors explain a bit more why the overlap of the vegetation season months was taken? Usually, this is from September of the previous year to August of the current one, or the whole hydrological year. In this study from May of the previous year to September of the current one. The correlations will be higher in this case, if the normal period, and therefore, can be overestimated.

Table 4. Negative correlations with seasonal temperature are unexpected and the mechanisms behind them should be discussed. I would really want to see if these correlations will remain if take a different window from September previous year to the August current one. Please check.

L. 431-433 in the methods mentioned that climatic analysis was performed for the period from May of the previous year to September of the current one. Did I miss something?

In general, it is difficult to follow for which periods correlation analysis was performed. In Table 4 (1975-2021), data are available from 1973 to 2018 (Fig. 1). Table 3 (1975-2018). Please define a common period for all calculations.

L. 531 seasonal temperatures and annual temperatures are replaced with seasonal and annual temperatures.

Author Response

Comments and Suggestions for Authors

Review of the article: forests-2592070 entitled Different adaptive potential of Norway spruce ecotypes in response to climate change in Czech long-term lowland experiment

The Authors evaluated the adaptive potential of Norway spruce trees from the Czech Republic in response to climate change by applying dendrometric inventory, dendrochronological sampling, and genetic analyses.

The Authors revealed negative correlations between tree growth and air temperature from three ecotypes (low, middle and high elevations). This is interesting funding and the period and months should be mentioned in the abstract as well as possible reasons for such correlations.

First, we would like to sincerely thank the reviewer for his professional comments that helped us significantly improve this study. We tried as much as possible to incorporate the given comments into the manuscript with maximum compliance with these comments.

Answer: We rewrited and added a better description on temperature in the abstract: All ecotypes exhibited a noteworthy negative correlation between tree-ring growth and seasonal temperature, annual temperature, temperature from previous year September to current year August, temperature from current June to current July, as well as individual monthly temperatures from previous May to current August.

The Authors found that the 3 studies' ecotypes significantly differ from each other. The Conclusion section looks like a discussion and should be shortened. Please provide the main conclusion and outcome of the study.

Answer: The conclusion was shortened and rewritten to remove the discussion phrases.

Figures: should be revised according to the Journal guidelines.

The paper can be of potential interest to the Forests Journal readers, but in its current form should be restructured and validated for the time periods and analyses used. My suggestions and details are below.

Abstract: Please refer to a period for which correlations with climatic parameters were revealed.

Answer: Done.

Please indicate the age of the trees used for analysis already in the abstract.

Answer: Done.

In general, the abstract should be shortened. Abbreviations for ecotypes lo elevation (EL), medium- elevation (ME) and high-elevation (HE) could be helpful.

Answer: The abstract was shortened and abbreviations were added to the whole manuscript.

Introduction

It is mentioned that around the 1970s to 1990s air pollution calamity.. But it is a bit unclear from which source the SO2 come from? Industry near by? Please clarify in one sentence before.

Answer: The pollution calamity was caused by coal power plants. It was added to the text.

61-95 mixture of study site description. Can be separated into intro and materials and methods parts?

Answer: This part of the introduction may be seen as similar to the site description but, this is the real origin of studied ecotypes and their morphologies. This description is used from the common knowledge about ecotypes. We would like to keep this in the introduction because it does not describe direct the study or any methodological methods in the manuscript. We have to also mention that sample collection for this experiment did not cover all attitudes mentioned in this part of the introduction. We have to mention that these altitudes for ecotypes are generally accepted for the origin of these ecotypes but the founders of the experiment did not have resources and capabilities to collect large sample size from all altitudes.

Minor comments

41 replace is in CR 48.1 to is 48.1% in CR.

Answer: Done.

L.47, L. 63,  please provide the citation in [].

Answer: Done.

56 either CR or the full name everywhere

Answer: Done.

Material and Methods

141 it is unclear what was collected. Please clarify about sampling and sampling design.

Answer: The seed material was collected. We clarified the sampling.

2.3 Research plot description

165 it is unclear about which research plot the Authors talk about. Please clarify. Is it related to the common trial above or it is different? It is very confusing. It might be helpful to clarify the following:

-dendrometric inventory

- dendrochronological sampling

- genetic analysis

Please describe study sites accordingly and the plant material collected for each analysis and ecotype.

Answer: The description of studie area was improved. “This experimental research plot is situated in one single studied lokality in a low-relief area with an altitude ranging from 320 to 340 m a.s.l. It was established in 1970 [50]. All three studied ecotypes are located in the studied research plot, where individual trees are randomly distributed over the area. Basic dendromegtric data were measured on each tree in the research area. Each tree was drilled for denrochonological analysis, with all assessed trees being recorded in terms of their origin and distribution across the area.“2.4 Data collection topic is incorrect here. “Data collection” – if the Authors reuse data or collect processed data for their study. My suggestion:  Sampling if the Authors collected samples for dendrochronological analysis or “Dendrochronological sampling”.

The number of samples is described in “Data collection” where is more detailed dendrochronology sampling.

However we think that separation of the big section data analysis will be better. We added undersections in data analysis for better reader overview:  :

2.5.1 Dendrocrhonological data processing and analysis

2.5.2 Tree-ring analysis with precitpitation and temperature

2.5.3 Heritability data processing

2.5.4 Stand structure and Biomass analysis

175 increment borer

Answer: Done.

253 “…monthly precipitation and temperature was performed for the period from May to September”. Could the Authors explain why they selected this seasonal window? Why not September of the previous year to August of the current one? Otherwise, there is an overlap that can result in unreasonable correlations or even overestimated correlations.

Answer: We responded in following way “This yearly and seasonal window was chosen for the evaluation of direct effect on studied ecotypes in current year“. The previous yearly effects cover DedroClim “This software covers individual months in previous vegetation season for monthly temperatures and monthly precipitation.” We also added new type of precipitation data from previous year September to current year August and data from current June to current July

The overlap of correlations is minimal, where monthly correlations, seasonal and annual correlations can describe closer differences in the effect of precipitation and temperature on individual ecotypes. In addition, it was mentioned in the introduction of the article that individual ecotypes show differences in morphology and germination time, which may result in different effects on the growth of ecotypes. We tried to describe everything in the discussion, where we also argue that HE spruce is the worst option, which can lead to germination time and differences in morphology.

257-259 Could the Authors explain a bit more why the overlap of the vegetation season months was taken? Usually, this is from September of the previous year to August of the current one, or the whole hydrological year. In this study from May of the previous year to September of the current one. The correlations will be higher in this case, if the normal period, and therefore, can be overestimated.

Answer? We added also this explanation to the paragraph „The monthly, seasonal, and annual correlations can describe closer differences in the effect of precipitation and temperature on individual ecotypes.“ and also The data of precipitation and temperature from previous year September to current year August and from current June to current July were applied to describe precipita-tion and temperature cumulative effect on the spruce ecotypes.

Table 4. Negative correlations with seasonal temperature are unexpected and the mechanisms behind them should be discussed. I would really want to see if these correlations will remain if take a different window from September previous year to the August current one. Please check.

Answer: These precipitation and temperature windows were added to the study. The correlation for temperature is still significantly negative. However, there were more significant results for precipitation. The effect of temperature is negative for this study area because spruce in the lowlands in Czech Republic does not prosper in the environment of higher temperatures but with limited precipitation, where low precipitation events are a more serious danger for Norway spruce. We discuss this effect more in the discussion section.

431-433 in the methods mentioned that climatic analysis was performed for the period from May of the previous year to September of the current one. Did I miss something?

Answer: Yes, the DencroClim calculation correlation window for monthly precipitation and temperature was defined from previous May to current September.

In general, it is difficult to follow for which periods correlation analysis was performed. In Table 4 (1975-2021), data are available from 1973 to 2018 (Fig. 1). Table 3 (1975-2018). Please define a common period for all calculations.

Answer: This time window was corrected to 1973 to 2018.

531 seasonal temperatures and annual temperatures are replaced with seasonal and annual temperatures.

Answer: Done.

Reviewer 2 Report

In the presented article the Authors related to the issue of adaptive potential of Norway spruce ecotypes to climate change in long-term lowland experiment

The manuscript itself is quite interesting although the drawn conclusions are rather obvious...as stated in last 8 lines of the Abstract and correspondingly in the Conclusions. This actually means the medium-elevation ecotype showed the greatest ring-width increment in spruce, the high-elevation ecotype showed a similar decline, failing to take advantage of growing conditions as effectively as the other two ecotypes. It is also a relatively most old study. However I still believe that it is valuable enough to be published.

There are other issues which have to be clarified.

Abstract:

The content in the abstract is too repetitive and cumbersome, please refine it, such as: As a result of climate change, Norway spruce (Picea abies [L.] Karst.) is dying across Europe, causing a large-scale disturbance that not only degrades the environment (soil erosion and 13 loss of water retention) but also causes a significant economic loss……This study deals with the 15 growth and genetics of the ecotypes of Norway spruce, the most important tree species in the lower 16 altitudes of the Czech Republic.

The result of the abstract was also more repetitive, please refine it. Such as: The medium-elevation ecotype appeared to be the most productive….

Introduction: 

There are some repetitions and overlaps in the first and second paragraphs of the introduction, and there is no clear division in terms of the serious impact of climate change on the tree species and the current ecotype status and existing problems of the tree species.

Almost the same problem exists in the next two paragraphs, please merge and streamline these two paragraphs:

Paragraph that the upland (medium-elevation) ecotype of spruce is adapted to a shorter vegetation 85 season when compared to the lowland ecotype….

and the Paragraph the high-elevation spruce ecotype has the shortest vegetation season compared to 90 the lowland and mountain spruce ecotypes….

Methodology:

The explanation of the study materials in the manuscript was not clear,

The basic information of materials collected from three altitudes is unclearHow was the planting density, diameter at breast height, and tree height growth of materials collected from three altitudes planted at the same location? How are the random effects of clones reflected in a mixed model? Were they clones of different lineages? But I don't know the situation of clone in the text, such as, how many clones? how many repetitions? and so on, need to supplement relevant explanations.

About the section 2.5. Data analysis, Many very detailed knowledge points are common sense to readers and can be streamlined appropriately.

Such as, Dendrochronological data from the Norway spruce ecotypes were processed with R software using the “dplR” and “pointRes” packages [61–64]. Negatively exponential detrending with an interleaved 67% spline was used to detrend individual trees, following the instructions in the “dplR” package [65]. Detrending serves to remove the age trend while preserving low-frequency growth signals [66,67]. An expressed population signal (EPS)…..

Line 274 What does HDR mean?

Figure 2. I suggest comparing the resistance, resilience, and resilience of high, medium, and low altitude spruce in one graph to more intuitively identify the differences in the three indicators of high, medium, and low altitude.

Results:

The sentence: Overall, the recovery index responds with a positive trend for the……

Maybe it’s belong to the section of discussion.

some same point was in the other part of the result section.

Conclusion:

About: It follows from the previous that in the silviculture of Norway spruce, it is necessary to carry out a consistent differentiation of spruce ecotypes in given habitat conditions, with an emphasis on the formation of stand mixtures in low-elevation and medium-elevation ecotypes to significantly increase their resistance potential under conditions of advancing climatic change.

This paragraph does not match the meaning of the sentence in the abstract that: In addition, this study also documented intra-population genetic variation within years of low growth, as evidenced by significant clonal heritability, or the main idea of that paragraph is not reflected in the conclusion.

Some parts of the conclusion need to be refined.

Line 236-237: Next, we applied the calculation of functions: resilience components resistance, recovery, resilience and relative resilience by [74].  Need to be polish.

Author Response

Comments and Suggestions for Authors

In the presented article the Authors related to the issue of adaptive potential of Norway spruce ecotypes to climate change in long-term lowland experiment

The manuscript itself is quite interesting although the drawn conclusions are rather obvious...as stated in last 8 lines of the Abstract and correspondingly in the Conclusions. This actually means the medium-elevation ecotype showed the greatest ring-width increment in spruce, the high-elevation ecotype showed a similar decline, failing to take advantage of growing conditions as effectively as the other two ecotypes. It is also a relatively most old study. However I still believe that it is valuable enough to be published.

There are other issues which have to be clarified.

Abstract:

The content in the abstract is too repetitive and cumbersome, please refine it, such as: As a result of climate change, Norway spruce (Picea abies [L.] Karst.) is dying across Europe, causing a large-scale disturbance that not only degrades the environment (soil erosion and 13 loss of water retention) but also causes a significant economic loss……This study deals with the 15 growth and genetics of the ecotypes of Norway spruce, the most important tree species in the lower 16 altitudes of the Czech Republic.

The result of the abstract was also more repetitive, please refine it. Such as: The medium-elevation ecotype appeared to be the most productive….

First, we would like to sincerely thank the reviewer for his professional comments that helped us significantly improve this study. We tried as much as possible to incorporate the given comments into the manuscript with maximum compliance with these comments.

Answer: We removed and rewrote the abstract to be less repetitive.

Introduction: 

There are some repetitions and overlaps in the first and second paragraphs of the introduction, and there is no clear division in terms of the serious impact of climate change on the tree species and the current ecotype status and existing problems of the tree species.

Answer: The paragraphs were rewritten.

Almost the same problem exists in the next two paragraphs, please merge and streamline these two paragraphs:

Paragraph that the upland (medium-elevation) ecotype of spruce is adapted to a shorter vegetation 85 season when compared to the lowland ecotype….

and the Paragraph the high-elevation spruce ecotype has the shortest vegetation season compared to 90 the lowland and mountain spruce ecotypes….

Answer: The vegetation season was for ecotypes was improved.

Methodology:

The explanation of the study materials in the manuscript was not clear,

The basic information of materials collected from three altitudes is unclear,How was the planting density, diameter at breast height, and tree height growth of materials collected from three altitudes planted at the same location? How are the random effects of clones reflected in a mixed model? Were they clones of different lineages? But I don't know the situation of clone in the text, such as, how many clones? how many repetitions? and so on, need to supplement relevant explanations.

Answer: We thank opponent for mentioning this missing information, plant material section was updated accordingly. Grafted trees were planted as clonal rows with ten individuals per clone in 3 m distances; distances between rows were 6 m. The whole site was subsequently pruned so that the final spacing is 6 m × 6 m. The average tree height on the plot was 20.6 m; sd = 2.95 m, and the average DBH was 33.1 cm; sd = 7.7 cm. Random effect of clone was accounted as term Z with the Multivariate Normal distribution (MVN), within the LMM model.

About the section 2.5. Data analysis, Many very detailed knowledge points are common sense to readers and can be streamlined appropriately.

Such as, Dendrochronological data from the Norway spruce ecotypes were processed with R software using the “dplR” and “pointRes” packages [61–64]. Negatively exponential detrending with an interleaved 67% spline was used to detrend individual trees, following the instructions in the “dplR” package [65]. Detrending serves to remove the age trend while preserving low-frequency growth signals [66,67]. An expressed population signal (EPS)…..

We made better separation of the big section Data analysis- We believe that it will be better to organize information for the reader. We added undersections in data analysis for better reader overview:  :

2.5.1 Dendrochronological data processing and analysis

2.5.2 Tree-ring analysis with precipitation and temperature

2.5.3 Heritability data processing

2.5.4 Stand structure and biomass analysis

Line 274 What does HDR mean?

Answer: It was explained by „HDR-height to DBH ratio“

Figure 2. I suggest comparing the resistance, resilience, and resilience of high, medium, and low altitude spruce in one graph to more intuitively identify the differences in the three indicators of high, medium, and low altitude.

Answer: We tried to explain these figures more intuitively way but it was near to the discussion. With all do respect to the reviewer. We would like to keep this description in this current form to avoid to avoid comparability with discussion.

Results:

The sentence: Overall, the recovery index responds with a positive trend for the……

Maybe it’s belong to the section of discussion.

some same point was in the other part of the result section.

Answer: This sentence was rewritten.

Conclusion:

About: It follows from the previous that in the silviculture of Norway spruce, it is necessary to carry out a consistent differentiation of spruce ecotypes in given habitat conditions, with an emphasis on the formation of stand mixtures in low-elevation and medium-elevation ecotypes to significantly increase their resistance potential under conditions of advancing climatic change.

This paragraph does not match the meaning of the sentence in the abstract that: In addition, this study also documented intra-population genetic variation within years of low growth, as evidenced by significant clonal heritability, or the main idea of that paragraph is not reflected in the conclusion.

Answer: We removed these assumptions sentences from the results in order to simplify and shorten the conclusion of this study, so that it better conveys the results of this work.

Some parts of the conclusion need to be refined.

Line 236-237: Next, we applied the calculation of functions: resilience components resistance, recovery, resilience and relative resilience by [74].  Need to be polish.

Answer: The sentence was rewritten.

Reviewer 3 Report

General comments

I reviewed with great interest the manuscript entitled “Different adaptive potential of Norway spruce ecotypes in response to climate change in Czech long-term lowland experimentsubmitted to the journal Forests.

The topic and the results are very interesting. However, though I am not a native English speaker, I believe that the manuscript needs revision for language and grammar, and should be revised, if possible, by a native speaker.

The paper is well-structured, the methods well-explained, and the discussion well-documented, though, some paragraphs should be moved from the Results to the Discussion.

However, I believe that the importance of the precipitation on growth would be better assessed through its cumulative effect, expressed in groupings of more than one month, instead of a correlation analysis with the effect of single months. The program Searscorr, for example, provides these best groupings.  

 Specific comments are provided in the enclosed PDF.

The manuscript needs revision for language.

Author Response

First, we want to sincerely thank the reviewer for a very professional review that helped us find many errors and inaccuracies in our study. Since there were many comments in the pdf of the manuscript, we responded only to the most important comments that we needed to respond to. We inserted many minor comments without response such as editing the text and editing the meaning of the sentences directly into the manuscript. Additional revision of the English language of the manuscript was carried out by a native speaker.

Answer: We respond on reviewer's comment on sampling of two trees. „Total of 76 dendrochronological samples were taken. The number of samples 181 collected for each ecotype studied ranged from 20 to 31 individuals (Table 3).

The collected data were used to compare climatic effects on tree rings and we understand that for a similar study, two samples are taken. However, there were after the years not enough healthy trees to sample, therefore we decided to not damage the tree with two cores. So, we decided that 1 sample could be enough when the EPS would reach a significant level (which it did). The age of the samples also indicates that variability in the tree ring growth is minimal, while SNR is higher than 11 (for the worst HE ecotype) . In general, we can argue that tree-ring growth has minimal signal noise.

Answer on detrending: We improved description of why we used this detrending. “Detrending serves to remove the age trend while preserving low-frequency growth signals and eliminates the effect of highly variable juvenile rings“.

Answer: We have to respond to software Seascorr which is part of the MATLAB program. Our department does not keep the license to use this software now and the other co-authors do not have this licensed program. Therefore, we cannot make calculations for Seascorr. However, we correlated some other precipitation and temperature windows to show cumulative effects. We made the additional correlations for temperature and precipitation from September to the current year August and from current June to the current July. The results of these new correlations are inserted into the study. These correlations show a wider perspective that is obtained from the results for each ecotype.

Answer: response on “The highest heritability value (0.7) ever (for all years observed), with the lowest estimation error, was observed in the negative pointer year 2013.” We ague that “This high value of heritability is caused by the high variability in the tree rings between the provenances of clones and ecotypes, when in this period the dieback effect on spruce stands is also significantly manifested.”

Answer: We answer on reviewer’s comment in the conclusion “The climate-growth relationships are investigated taking in consideration the correlations of single months. Another analysis considering the best groupings (months) would provide different results, as the effect of precipitation on growth is cumulative: single months P (example: March, April, and May, and June) can exhibit a non-significant correlation with growth, but a significant correlation if considered together (example: March-June P).

This will show better how precipitation control (or not) growth.

PS : Programs, like Seascorr. provides this best groupings.”

As we wrote before, we calculated the correlations for temperature and precipitation from September to the current year August, and from current June to the current July. Unfortunately, we do not have the license for Matlab software which is the greatest limitation for using the Seasscorr. However, we inserted two more windows of correlation for Table 4 of this study. The description of growth between studied ecotypes is one of the main aims of this article but we think that in our case Dendroclim analysis can be adequate in potential future comparison of possible signs in tree growth and morphology. We want to thank for improving our study and we consider your comments very valuable because we will use more different temperature and precipitation windows in the future for description more detailed effects on tree-ring growth.
